# Diagnostic performance of the TR Chagas Bio-Manguinhos rapid test for detecting anti-*Trypanosoma* cruzi IgG in human samples from three Southern Cone countries

Larissa de Carvalho Medrado Vasconcelos[1,2☉], Micaela Soledad Ossowski[3☉], Edimilson Domingos Silva[4], Daniel Dias Sampaio[2], Tycha Bianca Sabaini Pavan[1,2], Randrin Queiroz Viana Ferreira[1,2], Felipe Silva Santos de Jesus[1,2], Raúl Chadi[5], Marisa Liliana Fernández[6], Yolanda Hernández[6], Karina Andrea Gómez[3], Fred Luciano Neves Santos [1,2,7]*

**1** Advanced Public Health Laboratory (LASP), Gonçalo Moniz Institute, Oswaldo Cruz Foundation (FIOCRUZ-BA), Salvador, Bahia, Brazil, **2** Interdisciplinary Research Group in Biotechnology and Epidemiology of Infectious Diseases (GRUPIBE), Gonçalo Moniz Institute, Oswaldo Cruz Foundation (FIOCRUZ-BA), Salvador, Bahia, Brazil, **3** Instituto de Investigaciones en Ingeniería Genética y Biología Molecular "Dr. Héctor N. Torres" (INGEBI-CONICET), Buenos Aires, Argentina, **4** Immunobiological Technology Institute (Bio-Manguinhos), Oswaldo Cruz Foundation (FIOCRUZ-RJ), Rio de Janeiro, Rio de Janeiro, Brazil, **5** Hospital General de Agudos "Dr. Ignacio Pirovano", Buenos Aires, Argentina, **6** Instituto Nacional de Parasitología "Dr. Mario Fatala Chabén", Buenos Aires, Argentina, **7** Integrated Translational Program in Chagas Disease from FIOCRUZ (Fio-Chagas), Oswaldo Cruz Foundation (FIOCRUZ-RJ), Rio de Janeiro, Rio de Janeiro, Brazil

☉ These authors contributed equally to this work.
* fred.santos@fiocruz.br (FNLS)

## Abstract

Chagas disease (CD) remains a major public health challenge in Latin America, largely due to persistent underdiagnosis, particularly in endemic and resource-limited settings. Rapid diagnostic tests (RDTs) offer a pragmatic approach to expand sero-logical screening; however, their performance varies across epidemiological contexts and parasite genetic backgrounds. The TR Chagas Bio-Manguinhos test, based on recombinant chimeric antigens, has shown high sensitivity in previous evalua-tions, but data from Southern Cone remain limited. We conducted a cross-sectional diagnostic accuracy study using 311 anonymized human plasma samples obtained in Argentina, including 233 *Trypanosoma cruzi*–positive and 78 negative samples classified by a composite reference standard based on at least two independent serological assays. Diagnostic performance of the TR Chagas Bio-Manguinhos rapid test was assessed in terms of sensitivity, specificity, accuracy, likelihood ratios, diagnostic odds ratio, and agreement. Sensitivity was further stratified by country of origin (Argentina, Bolivia, Paraguay) and by cardiac involvement according to the Kuschnir classification. The TR Chagas Bio-Manguinhos test demonstrated high

**Data availability statement:** All relevant data are in the manuscript and its supporting information files.

**Funding:** This study was supported by the Coordination for the Improvement of Higher Education Personnel (CAPES, Brazil; Finance Code 001 to FLNS, 88887.896806/2023-00 to LCMV, and 88887.115606/2025-00 to FSSJ), INOVA Fiocruz – Clinical Research in Chagas Disease (VPPCB-001-FIO-22-2-11 to FLNS), the National Council for Scientific and Technological Development-Brazil (CNPq; 442269/2024-2 to FLNS, 422824/2025-9 to FLNS, and 121970/2024-8 to RQVF), and the Bahia Research Foundation (FAPESB; APP0007/2023 to FLNS). FLNS is a research fellow of the CNPq (306448/2023-8). The funders had no role in study design, data collection and analysis, decision to publish, or preparation of the manuscript.

**Competing interests:** The authors have declared that no competing interests exist.

overall performance, with a sensitivity of 97.0% (95% CI: 93.9–98.5%), specificity of 94.9% (95% CI: 87.5–98.0%), and accuracy of 96.5% (95% CI: 93.8–98.0%). Agreement with the reference standard was almost perfect ($\kappa = 0.91$). Sensitivity remained consistently high across geographical origins and Kuschnir cardiac stages, with no statistically significant differences between subgroups. The negative likelihood ratio (0.03) indicated strong ability to rule out infection. These findings provide the first independent evidence supporting the high performance of the TR Chagas Bio-Manguinhos test in people from Argentina, Bolivia and Paraguay. The combination of very high sensitivity and robust specificity supports its use as a frontline serological screening tool for chronic *T. cruzi* infection in the Southern Cone, provided that reactive results are subsequently confirmed by laboratory- based assays in accordance with international guidelines.

## Author summary

Chagas disease (CD) remains a major public health problem in Latin America, largely because many infected individuals are never diagnosed. This challenge is particularly pronounced in endemic and resource-limited settings, where access to laboratory-based serological tests is restricted. Rapid diagnostic tests offer an important opportunity to expand screening, but their performance can vary across populations and geographic regions, making independent validation essential. In this study, we evaluated the diagnostic performance of the TR Chagas Bio-Manguinhos rapid test using samples from individuals originating from Argentina, Bolivia, and Paraguay. Test results were compared with a reference standard based on multiple conventional serological assays. The rapid test demonstrated high sensitivity and specificity, with excellent agreement with the reference standard. Its performance remained consistently high across different countries of origin and across Kuschnir cardiac stages. These findings provide the first independent evidence supporting the reliability of the TR Chagas Bio-Manguinhos rapid test in Southern Cone populations and support its use as a frontline screening tool for chronic *T. cruzi* infection, provided that reactive results are confirmed by laboratory-based assays in accordance with international guidelines.

## Introduction

Chagas disease (CD), caused by the protozoan *Trypanosoma cruzi*, remains one of the most important parasitic diseases in Latin America despite major advances in vector control and public health interventions. Argentina is among the most heavily affected countries, with an estimated 706,112 individuals currently infected and approximately 6.2 million people (14% of the national population) living at risk of infection [1]. At the regional level, the Southern Cone accounts for nearly 78% of all global CD cases, contributing to an estimated 4.9 million infections worldwide [1]. More than a century after its discovery, CD continues to face critical barriers to early

detection and effective treatment, driven by persistent vectorial and congenital transmission and by the large proportion of chronically infected individuals who remain undiagnosed [2–4].

Early serological diagnosis is essential to expand access to timely etiological treatment, yet the diagnostic landscape for CD is operationally and technically complex. No single assay is recognized as a gold-standard; instead, current guidelines recommend the use of at least two serological tests based on different methodological principles to confirm *T. cruzi* infection [5,6]. These assays typically require laboratory infrastructure, specialized equipment, and trained personnel. When results are discordant, a third confirmatory test is needed. Although this multistep algorithm is reliable, it is difficult to implement in remote, resource-limited, or highly endemic settings, where diagnostic capacity is constrained, and health systems already face substantial logistical challenges [4].

Rapid diagnostic tests (RDTs) represent a promising strategy to expand access to screening for *T. cruzi* infection. Their operational simplicity, portability, and rapid turnaround time facilitate deployment in primary healthcare facilities, community-based screening initiatives, and outreach activities. However, the diagnostic performance of commercially available RDTs has been heterogeneous, and only a subset of assays has undergone rigorous evaluation across distinct epidemiological contexts or *T. cruzi* discrete typing units (DTUs) [7–10]. The TR Chagas Bio-Manguinhos test, developed by Bio-Manguinhos/Fiocruz (Brazil), is an immunochromatographic lateral-flow assay (LFA) that detects anti-*T. cruzi* IgG using two recombinant chimeric antigens, providing results within 15 minutes [11]. Preliminary evaluations conducted in Brazil [7], Bolivia [9], and Colombia [8] have shown high sensitivity, but its performance in Southern Cone countries, where distinct DTUs predominate, has not yet been fully assessed. Because antigenic diversity and host immune responses vary geographically, external validation in populations from neighboring countries is essential before broader regional implementation.

In this context, we evaluated the diagnostic performance of the TR Chagas Bio-Manguinhos test using plasma samples from Argentina, Bolivia and Paraguay, applying international guidelines for diagnostic test validation, and additionally explored whether test sensitivity varied according to country of origin and cardiac involvement.

## Methods

### Ethics statement

This study was approved by the Institutional Review Boards (IRB) of the Instituto Nacional de Parasitología "Dr. Mario Fatala Chabén" (Study Protocols Nº 3–2018, 19–2019, and 1-2020-2024) and the Hospital General de Agudos "Dr. Ignacio Pirovano" (Study Protocol Nº 56–2015), Buenos Aires, Argentina. Written informed consent was obtained from all participants at the time of sample collection.

### Study design and samples

This retrospective observational, cross-sectional diagnostic performance study used anonymized human plasma obtained from the INGEBI-CONICET serotheque/biorepository in Buenos Aires, Argentina. Although samples were obtained from individuals originating from Argentina, Bolivia, and Paraguay, this was not a multicenter study, but rather a single-center evaluation based on samples analyzed in Buenos Aires, Argentina. Samples were obtained from individuals evaluated at the Instituto Nacional de Parasitología "Dr. Mario Fatala Chabén" and at the Hospital General de Agudos "Dr. Ignacio Pirovano," both in Buenos Aires, Argentina, between 2015 and 2024. Samples were collected in the participating institutions either during routine clinical evaluation or during scheduled blood collection visits requested as part of the diagnostic workup, and were locally processed to obtain plasma for storage in the repository. Plasma aliquots were stored at –20 °C in multiple tubes to minimize repeated freeze–thaw cycles. Because this was a retrospective study based on archived material collected over several years, storage duration varied according to the date of collection.

Sample size calculations were performed assuming a 1.7% margin of error (half-width of the 95% confidence interval), expected sensitivity of 99%, specificity of 99.5%, and a 95% confidence level, based on standard formulas for proportions

in diagnostic accuracy studies [12], resulting in a minimum requirement of 132 *T. cruzi*–positive and 67 negative plasma samples. The study included 233 positive and 78 negative plasma samples from individuals originating from Argentina, Bolivia, and Paraguay. Although city- or province-level origin was unavailable, all participants resided in Buenos Aires at the time of medical evaluation. Samples were selected as convenience samples from the repository according to availability and eligibility criteria; therefore, the possibility of spectrum bias cannot be excluded. No specific sample size was prospectively defined for stratified analyses by geographical origin or cardiac involvement; therefore, these subgroup analyses should be interpreted as exploratory.

Samples were classified as *T. cruzi*–positive or –negative based on the original serological results obtained at the participating institutions using commercial assays routinely employed for the diagnosis of chronic *T. cruzi* infection, following international recommendations [5]. At the Instituto Nacional de Parasitología "Dr. Mario Fatala Chabén", the diagnostic workup included CMIA Architect Chagas (ARCHITECT Chagas, Abbott, Illinois, USA) and Chagatest ELISA recombinante v.4.0 (Wiener lab, Rosario, Argentina). At the Hospital General de Agudos "Dr. Ignacio Pirovano", routine diagnosis included Chagatest HAI (Wiener lab, Rosario, Argentina) and CMIA on the Alinity i platform (Abbott, Illinois, USA). In cases of discrepancy, an additional hemagglutination assay (SERODIA-CHAGAS, Fujirebio Inc., Tokyo, Japan) was used in routine practice; however, discordant cases were not included in the sample panel analyzed in the present study because the aim was to evaluate the index test in a panel with a defined serological classification. The reference classification was based on the original serological results obtained in the clinical setting and was not re-established by repeating all reference assays specifically for the present study. In contrast, the TR Chagas Bio-Manguinhos assay was performed later on archived plasma samples. Because this was a retrospective study based on repository material, the time interval between the original reference testing and the index test was not standardized and may have varied across samples. In addition, samples were re-evaluated in our laboratory using an in-house ELISA based on *T. cruzi* lysate as an additional internal characterization step, following a previously described protocol [13]. Detailed serological profiles are provided in S1 Table.

Positive samples were obtained from individuals with chronic *T. cruzi* infection for whom epidemiological and clinical data were available, including cardiac involvement assessed using the Kuschnir staging system [14]. This classification includes four stages: stage 0, defined by reactive serology with a normal electrocardiogram and normal chest X-ray or echocardiogram without evidence of left ventricular dilatation; stage 1, defined by reactive serology and an abnormal electrocardiogram but with normal chest X-ray or echocardiogram and absence of left ventricular dilatation; stage 2, characterized by reactive serology, abnormal electrocardiogram, and radiological or echocardiographic evidence of left ventricular dilatation without clinical or radiological signs of heart failure; and stage 3, representing the most advanced form, defined by reactive serology, abnormal electrocardiogram, radiological or echocardiographic evidence of left ventricular dilatation, and the presence of heart failure. Cardiac evaluations were conducted at Hospital General de Agudos "Dr. Ignacio Pirovano" and Instituto Nacional de Parasitología "Dr. Mario Fatala Chabén".

### TR Chagas Bio-Manguinhos

The index test was the TR Chagas Bio-Manguinhos (Bio-Manguinhos/Fiocruz, Rio de Janeiro, Brazil), a lateral-flow immunochromatographic assay for the qualitative detection of anti-*T. cruzi* IgG. The device incorporates two recombinant chimeric antigens (IBMP-8.1 and IBMP-8.4) and an internal control line immobilized on a multimembrane strip assembled under an adhesive card to permit capillary flow [11]. According to the manufacturer's instructions, five microliters of plasma were applied to the sample port, followed by three drops of running buffer. Results were interpreted visually after 15 minutes at room temperature (≈22°C). A positive result was defined by the presence of any test line (IBMP-8.1 and/or IBMP-8.4) together with the control line, whereas negative results showed only the control line.

Two readers independently interpreted the test bands and were blinded to the reference serological results and to all clinical and epidemiological information associated with the samples. Readings were performed independently, without access to each other's interpretations. In cases of uncertainty or disagreement, a third observer adjudicated the result by

direct visual inspection of the original test cassette. All readings were performed by laboratory personnel under routine laboratory conditions and in accordance with the manufacturer's instructions. Tests lacking a visible control line were classified as invalid. Photographic documentation of test results was obtained during the study and is available upon reasonable request.

## Data Analysis

Diagnostic performance metrics included sensitivity, specificity, accuracy, positive and negative likelihood ratios, and the diagnostic odds ratio (DOR), all calculated from 2×2 contingency tables [15]. Ninety-five percent confidence intervals (95% CI) were estimated using exact binomial methods. Agreement between the index test and the reference standard was assessed using Cohen's kappa coefficient and interpreted according to established thresholds: poor ($\kappa = 0$), slight ($0 < \kappa \leq 0.20$), fair ($0.21 < \kappa \leq 0.40$), moderate ($0.41 < \kappa \leq 0.60$), substantial ($0.61 < \kappa \leq 0.80$), and almost perfect ($0.81 < \kappa \leq 1.0$) [16].

To explore potential variations in diagnostic performance across subgroups, sensitivity was estimated according to geographical origin (Argentina, Bolivia, and Paraguay) and Kuschnir cardiac stage. Pairwise differences in sensitivity were evaluated using Fisher's exact test. Because no specific sample size was prospectively defined for subgroup analyses, these comparisons were considered exploratory.

Age was summarized as median and interquartile range (IQR). Categorical variables, including sex, country of origin, and serological classification (*T. cruzi*–positive or –negative), were described using absolute and relative frequencies. A study flowchart (Fig 1) was prepared in accordance with the Standards for Reporting of Diagnostic Accuracy Studies (STARD) guidelines [17]. All diagnostic performance calculations and statistical analyses were performed using MedCalc for Windows v.20.190 (MedCalc Software, Ostend, Belgium).

## Results

### Samples characterization

A total of 311 previously collected, anonymized human plasma samples were included in the study, comprising individuals originating from Argentina (n=247; 79.4%), Bolivia (n=43; 13.8%), and Paraguay (n=21; 6.8%), all residing in Buenos Aires at the time of clinical evaluation. Of these, 233 samples were classified as *T. cruzi*–positive and 78 as *T. cruzi*–negative according to the reference standard.

Individuals classified as *T. cruzi*–positive had a median age of 56 years (IQR: 50–63 years) and a female-to-male ratio of 1.1:1, whereas those classified as *T. cruzi*–negative had a median age of 41.4 years (IQR: 29.3–50 years) and a female-to-male ratio of 0.83:1. Among infected individuals, cardiac involvement according to the Kuschnir classification was distributed as follows: stage 0, 102 (43.8%); stage 1, 83 (35.6%); stage 2, 8 (3.4%); and stage 3, 24 (10.3%). Kuschnir data were unavailable for 16 individuals (6.9%). Gastrointestinal involvement was not investigated.

### Diagnostic performance of TR Chagas Bio-Manguinhos

The TR Chagas Bio-Manguinhos rapid test showed high diagnostic performance when evaluated against the reference standard (Table 1). Among the 233 *T. cruzi*–positive samples, 226 were correctly identified, resulting in a sensitivity of 97.0% (95% CI: 93.9%–98.5%). Of the 78 negative samples, 74 were correctly classified, corresponding to a specificity of 94.9% (95% CI: 87.5%–98.0%). Overall accuracy was 96.5% (95% CI: 93.8%–98.0%), indicating strong discrimination between infected and non-infected individuals.

The positive likelihood ratio was 18.9 (95% CI: 7.3–49.1), whereas the negative likelihood ratio was 0.03 (95% CI: 0.02–0.07), supporting the ability of the assay to rule in and, particularly, to rule out infection. The diagnostic odds ratio (DOR) was 597.3 (95% CI: 170.1–2,097.8), and agreement between the index test and the reference standard was almost perfect (Table 1), with a Cohen's kappa of 0.91 (95% CI: 0.85–0.96). Inter-reader agreement was high, with only one test requiring adjudication by a third observer. No invalid test results were observed.

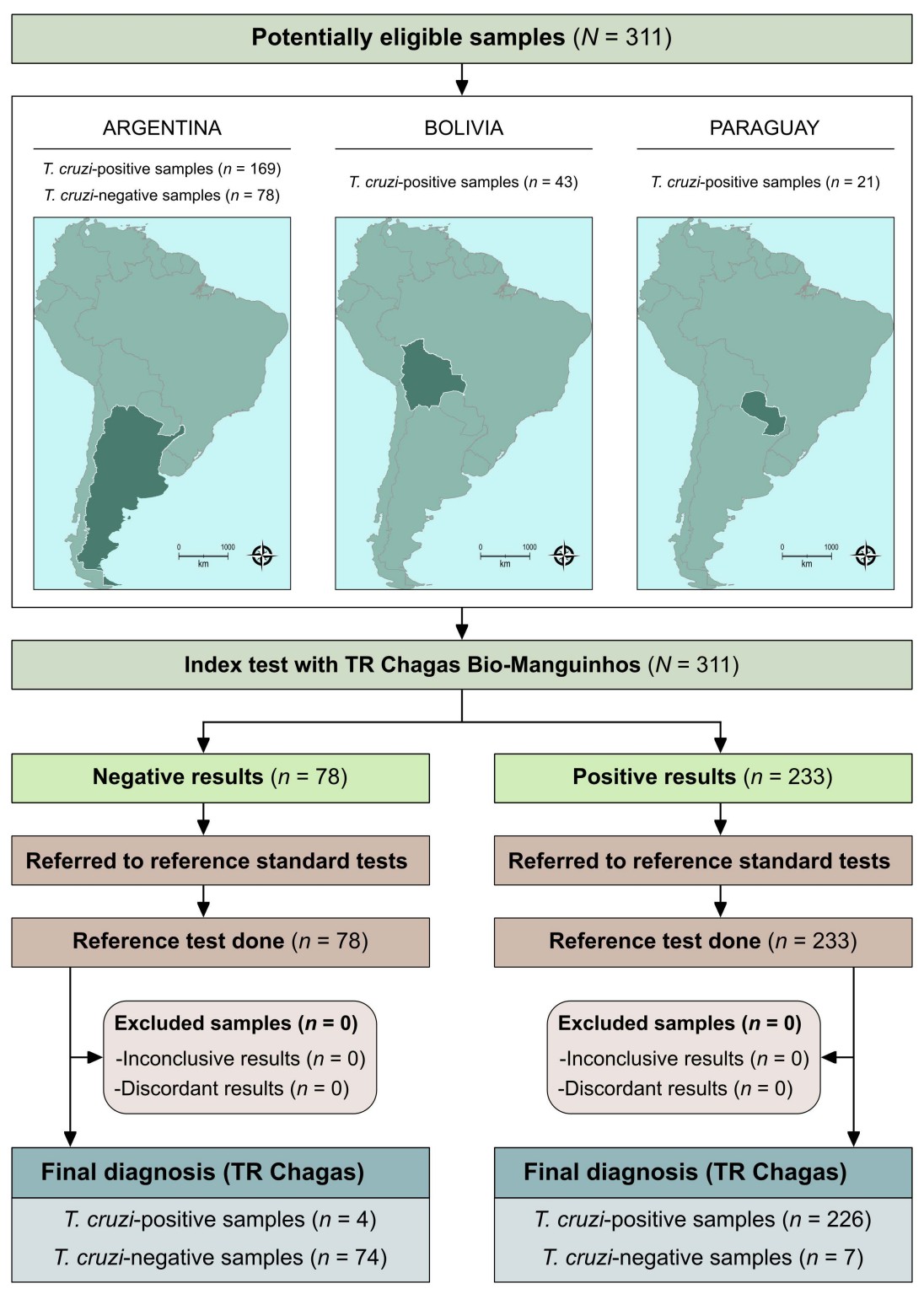

**Fig 1. Flowchart of the study design in accordance with the Standards for Reporting of Diagnostic Accuracy Studies (STARD) guidelines.**
The base map was obtained from the geoBoundaries cartographic database in shapefile format (.shp) (www.geoboundaries.org), an open database of political administrative boundaries distributed under the CC BY 4.0 license, and was subsequently reformatted and analyzed using QGIS version 3.10 (Geographic Information System, Open-Source Geospatial Foundation Project. https://qgis.org).

**Table 1. Diagnostic performance and agreement of TR Chagas Bio-Manguinhos for the detection of *Trypanosoma cruzi* IgG, using the original concordant serological classification obtained at the participating institutions as the reference standard.**

| Metric | Estimate | 95% CI |
|---|---|---|
| Sensitivity % | 97.0 | 93.9–98.5 |
| Specificity % | 94.9 | 87.5–98.0 |
| Accuracy % | 96.5 | 93.8–98.0 |
| Positive likelihood ratio | 18.9 | 7.3–49.1 |
| Negative likelihood ratio | 0.03 | 0.02–0.07 |
| Diagnostic odds ratio | 597.3 | 170.1–2,097.8 |
| Cohen's Kappa | 0.91 | 0.85–0.96 |

CI (Confidence interval)

## Analysis of TR Chagas Bio-Manguinhos sensitivity by geographical origin

To assess whether sensitivity varied according to geographical origin, it was estimated separately for individuals born in Argentina, Bolivia, and Paraguay (Table 2). Specificity was not stratified by geographical origin because all negative samples originated from Argentina. Among Argentine-born participants, the test correctly identified 163 of 169 *T. cruzi*–positive individuals, resulting in a sensitivity of 96.4% (95% CI: 92.5%–98.4%). In the Bolivian-born group, 42 of 43 infected individuals were correctly identified, corresponding to a sensitivity of 97.7% (95% CI: 87.9%–99.6%). All 21 Paraguayan-born individuals with confirmed infection were correctly classified, yielding a sensitivity of 100% (95% CI: 84.5%–100%).

Pairwise comparisons using Fisher's exact test revealed no statistically significant differences in sensitivity between individuals from Argentina, Bolivia, and Paraguay (all p > 0.60). Although sensitivity was numerically highest among Paraguayan-born individuals, followed by Bolivian- and Argentine-born participants, these differences should be interpreted with caution given the small number of false-negative results and the limited sample size in some subgroups.

## Analysis of TR Chagas Bio-Manguinhos sensitivity by cardiac involvement

When stratified according to Kuschnir cardiac stage, the TR Chagas Bio-Manguinhos test showed high sensitivity across all clinical categories (Table 3). Specificity was not analyzed by Kuschnir stage because this classification was available only for individuals with confirmed *T. cruzi* infection. Sensitivity was highest in stage 0 (98.0%) and stage 1 (97.6%), indicating high diagnostic performance in individuals with normal or mildly abnormal cardiac function. Sensitivity was slightly lower in stage 2 (87.5%) and stage 3 (91.7%), although these values remained clinically acceptable and are likely influenced by the small number of samples in advanced stages and by biological variability associated with myocardial remodeling.

**Table 2. Sensitivity of TR Chagas Bio-Manguinhos according to geographical origin.**

| Geographical origin | Total (n) | TP | FN | Sensitivity % (95% CI) |
|---|---|---|---|---|
| Argentina | 169 | 163 | 6 | 96.4% (92.5%–98.4%) |
| Bolivia | 43 | 42 | 1 | 97.7% (87.9%–99.6%) |
| Paraguay | 21 | 21 | 0 | 100% (84.5%–100%) |

TP (True positives); FN (False negatives): CI (Confidence interval). Pairwise comparisons were performed using Fisher's exact test; no statistically significant differences were observed between geographical groups (all p > 0.05).

**Table 3. Sensitivity of TR Chagas Bio-Manguinhos according to Kuschnir cardiac stage.**

| Kuschnir stage | Total (n) | TP | FN | Sensitivity % (95% CI) |
|---|---|---|---|---|
| 0 | 102 | 100 | 2 | 98.0% (93.1%–99.5%) |
| 1 | 83 | 81 | 2 | 97.6% (91.6%–99.3%) |
| 2 | 8 | 7 | 1 | 87.5% (52.9%–97.8%) |
| 3 | 24 | 22 | 2 | 91.7% (74.2%–97.7%) |

TP (True positives); FN (False negatives): CI (Confidence interval). Pairwise comparisons were performed using Fisher's exact test; no statistically significant differences were observed between Kuschnir stages (all p > 0.05).

Pairwise comparisons of sensitivity across Kuschnir stages using Fisher's exact test showed no statistically significant differences between any stage comparison (all p > 0.16), indicating that the TR Chagas Bio-Manguinhos test maintains stable sensitivity across varying degrees of chronic Chagas cardiomyopathy. However, these findings should be interpreted with caution because the number of samples in advanced stages was limited, particularly in stage 2.

## Discussion

The present study provides the first independent evaluation of the TR Chagas Bio-Manguinhos rapid test using plasma samples from individuals originating from Argentina, Bolivia, and Paraguay. The assay demonstrated high diagnostic performance, with a sensitivity of 97.0% and a specificity of 94.9%, placing its performance within the upper range reported for lateral-flow assays (LFAs) used for the serological diagnosis of chronic *T. cruzi* infection in Latin America. These findings add to the growing evidence that rapid immunochromatographic assays incorporating next-generation recombinant chimeric antigens can support decentralized screening strategies, particularly in settings where access to conventional laboratory infrastructure remains limited [4].

Independent evaluations conducted in Bolivia, Colombia, Guatemala, and Brazil have shown substantial heterogeneity in the diagnostic performance of RDTs across brands, study designs, and epidemiological contexts. In Bolivia, sensitivities ranged from 62.2% to 97.7% and specificities from 78.6% to 100%, with only a subset of assays achieving simultaneously high sensitivity and specificity [9]. Similarly, laboratory-based assessments in Colombia showed that while some RDTs achieved specificity values approaching 100%, others performed substantially less well, underscoring marked brand-to-brand variability [8]. Field evaluations in Guatemala, further indicated that operational conditions may reduce sensitivity, highlighting the need for context-specific validation before implementation [10]. Against this heterogeneous background, the diagnostic sensitivity observed for the TR Chagas Bio-Manguinhos assay in the present study falls within the upper range reported for LFAs in previous evaluations.

A particularly relevant finding of the present study is the specificity observed for the TR Chagas Bio-Manguinhos test, which differed from values reported in previous independent evaluations conducted in Brazil, Colombia, and Bolivia. In the retrospective laboratory-based evaluation by Marchiol *et al*. [8], specificity varied widely across different RDT brands and, for some assays, fell below 90% even under controlled conditions. Likewise, in the Brazilian multicenter evaluation by Iturra *et al*. [7], although the TR Chagas Bio-Manguinhos test achieved maximal sensitivity, its specificity was substantially lower (approximately 78.5%) than that observed here. In Bolivia, López *et al*. [9] reported that even among the best-performing LFAs, specificity showed greater variability than sensitivity, reflecting important differences across study panels and epidemiological settings. The discrepancy between the specificity observed in the present study and that reported by López *et al*. [9], despite broadly similar sensitivity values, is likely explained by differences in study design, negative panel composition, and epidemiological context rather than by intrinsic differences in test specificityalone.

The higher specificity observed here is plausibly explained by methodological and panel-related factors rather than intrinsic differences in test behavior. As in the aforementioned studies, negative samples in the present work were classified using a stringent serological reference standard based on concordant results obtained with routine diagnostic assays performed at the participating institutions [5]. However, prior evaluations frequently included more heterogeneous negative panels [7–9]. In particular, Iturra et al. [7] included samples from individuals with other infectious diseases associated with potential serological cross-reactivity, such as leishmaniasis. Therefore, differences in negative panel composition across studies may have contributed to variation in specificity estimates. In addition, the smaller number of negative samples in the present study may have reduced the precision of specificity estimation relative to previous evaluations that included larger negative panels. Collectively, these observations underscore that both sensitivity and specificity in Chagas RDT evaluations are not immutable test properties, but rather context-dependent parameters influenced by study design, sample composition, parasite genetic diversity, host immune status, and the reference standard used for classification. Importantly, even within this variability, the specificity observed here remains compatible with the intended role of the TR Chagas Bio-Manguinhos test as a screening tool, provided that reactive results are systematically confirmed by laboratory-based assays, in accordance with PAHO [5] and WHO [18] recommendations.

Antigen selection remains a central determinant of assay performance in Chagas serology. Previous studies have emphasized that manufacturers often do not disclose the antigenic composition of commercial RDTs, and that differences in antigen source, structure, and production may contribute to variation in sensitivity and specificity [8,9]. The TR Chagas Bio-Manguinhos test incorporates two well-characterized recombinant chimeric antigens, IBMP-8.1 and IBMP-8.4, which have shown consistently high discriminatory performance across multiple platforms, including ELISA [19–24], LFA [6,7,11], Western blot [25] and biosensor-based approaches [26,27], as well as in different host species [28,29]. More detailed information on the specific peptide or epitope composition of the commercial assay is not publicly available, which limits direct comparison with other commercial platforms at a finer antigenic level. Nevertheless, the use of IBMP-8.1 and IBMP-8.4 likely contributes to the consistent sensitivity observed in the present cohort. Importantly, the test evaluated here corresponds to the same commercial version used in previous evaluations, with no changes in antigenic composition or test design, thereby supporting direct comparability across studies. Indeed, no significant differences in sensitivity were observed between individuals from Argentina, Bolivia, and Paraguay, despite known regional variation in immune responses and circulating parasite lineages.

The potential impact of *T. cruzi* DTU diversity on diagnostic accuracy remains a major concern in the serological diagnosis of chronic infection [30]. Bolivia is predominantly affected by TcI and TcV, whereas Argentina historically exhibits higher prevalence of TcII and TcVI. Despite these theoretical concerns, comparative studies increasingly suggest that well-designed recombinant chimeric antigens can detect antibody responses across multiple DTUs [21]. The Bolivian evaluation reported that several LFAs maintained high accuracy despite DTU variation [9], and multi-country analyses have shown similar performance when lineages TcI, TcII, and TcV are directly compared. Our findings are consistent with this interpretation, as sensitivity remained high across individuals originating from different Southern Cone countries [21,22]. However, these analyses should be interpreted cautiously, since country of origin was used as a proxy for epidemiological background and the study was not designed to directly evaluate DTU-specific performance.

An additional contribution of this study is the exploratory analysis of sensitivity according to Kuschnir cardiac stage. Sensitivity remained high in stages 0 and 1 and was somewhat lower in stages 2 and 3, although none of these differences reached statistical significance. Advanced Chagas cardiomyopathy is associated with immune remodeling and may affect antibody kinetics [31,32], which could partially contribute to this pattern. At the same time, the number of individuals in advanced stages, particularly stage 2, was small, resulting in wide confidence intervals and limited statistical power. Therefore, these findings should be interpreted as exploratory rather than as evidence of equivalent performance across all clinical stages.

RDTs have been repeatedly proposed as frontline tools to expand access to screening for *T. cruzi* infection in endemic regions and migrant populations [4]. WHO and PAHO emphasize their utility in decentralized settings and support their use as first-line seroepidemiological tools when adequately validated. In the present study, the high sensitivity observed across the overall sample minimizes the risk of false-negative classifications, a critical consideration given that underdiagnosis of *T. cruzi* infection remains one of the principal barriers to effective disease control. The very low negative likelihood ratio further supports the usefulness of a negative result for ruling out chronic infection in most epidemiological settings. Nevertheless, positive results should continue to be confirmed using laboratory-based assays, particularly in low-prevalence settings where even modest reductions in specificity can significantly affect positive predictive value.

From a programmatic perspective, large-scale screening strategies based on RDTs may generate substantial numbers of reactive results requiring downstream confirmatory testing, with direct implications for cost and operational feasibility. For this reason, formal cost-effectiveness analyses will be important to assess the feasibility of deploying this assay at scale in different epidemiological scenarios. Such analyses should consider not only the costs associated with additional confirmatory testing, but also the consequences of false-negative results, including delayed diagnosis, missed opportunities for treatment, and downstream costs related to progression to clinically significant disease. Future applied research should also investigate whether high-performing RDTs could contribute to simplified confirmatory strategies in specific contexts, particularly in underserved settings where access to conventional laboratory methods is limited. If such approaches prove analytically and programmatically viable, they could have broader public health implications by accelerating diagnosis, facilitating linkage to care, and expanding access to diagnostic services in remote and vulnerable populations.

This study has several strengths, including the use of a well-characterized plasma panel and classification based on concordant serological results from routine diagnostic practice, and a sample size exceeding minimum power requirements, enhancing the precision of performance estimates. Nevertheless, several limitations should be acknowledged. The number of negative samples was lower than ideal for high-precision estimation of specificity, which likely contributed to wider confidence intervals for this parameter. Because samples were selected as convenience samples from a repository rather than through consecutive recruitment, spectrum bias cannot be excluded. The retrospective design also precluded standardization of the time interval between the original reference testing and performance of the index test. Although samples were stored under controlled conditions, the potential effect of long-term storage on antibody stability cannot be completely excluded. In addition, province-level geographical data and information on gastrointestinal involvement were unavailable, limiting more granular epidemiological and clinical interpretation. An additional limitation concerns the exclusion of samples with discordant conventional serological results from the study panel. This decision was deliberate and consistent with the analytical objective of evaluating the index test in a well-characterized sample set with a defined serological classification, as recommended in diagnostic validation studies [15]. Excluding discordant samples reduces uncertainty in the reference classification and helps avoid distortion of performance estimates caused by samples with unresolved infection status. However, this approach may also have led to slightly optimistic estimates of diagnostic performance, particularly because diagnostically challenging samples that are more likely to occur in real-world screening algorithms were not represented. Consequently, the present findings should be interpreted primarily as evidence of analytical performance in a well-defined retrospective panel, and caution is warranted when extrapolating these estimates to broader programmatic settings in which inconclusive or discrepant serological profiles are part of routine practice. No formal external quality and lot-to-lot variation assessments were performed, and whole blood was not evaluated; therefore, further field-based studies remain warranted to better characterize performance under true point-of-care conditions.

Taken together with independent evaluations conducted in Brazil [7], Bolivia [9], Colombia [8], and Guatemala [10], the present findings strengthen the evidence that high-quality RDTs incorporating recombinant chimeric antigens can achieve diagnostic performance for screening of chronic *T. cruzi* infection. The TR Chagas Bio-Manguinhos rapid test showed high overall sensitivity and specificity in this study. Although the subgroup analyses suggested comparable performance

across countries of origin and Kuschnir stages, these results should be interpreted with caution because of the limited sample sizes and wide confidence intervals in some strata. Overall, the assay appears to be a reliable screening tool for expanded diagnostic strategies in the Southern Cone, provided that reactive results are incorporated into confirmatory algorithms in accordance with international recommendations.

## Supporting information

**S1 Table. Serological profiles of individuals included in the study.**
(XLSX)

## Acknowledgments

We acknowledge the Institute of Technology on Immunobiologicals (Bio-Manguinhos/Fiocruz) for providing the TR Chagas Bio-Manguinhos rapid tests used in this study.

## Author contributions

**Conceptualization:** Micaela Soledad Ossowski, Edimilson Domingos Silva, Raúl Chadi, Marisa Liliana Fernández, Yolanda Hernández, Karina Andrea Gómez, Fred Luciano Neves Santos.

**Data curation:** Marisa Liliana Fernández, Yolanda Hernández, Karina Andrea Gómez, Fred Luciano Neves Santos.

**Formal analysis:** Larissa de Carvalho Medrado Vasconcelos, Micaela Soledad Ossowski, Edimilson Domingos Silva, Daniel Dias Sampaio, Randrin Queiroz Viana Ferreira, Felipe Silva Santos de Jesus, Raúl Chadi, Marisa Liliana Fernández, Yolanda Hernández, Karina Andrea Gómez, Fred Luciano Neves Santos.

**Funding acquisition:** Karina Andrea Gómez, Fred Luciano Neves Santos.

**Investigation:** Micaela Soledad Ossowski, Tycha Bianca Sabaini Pavan, Randrin Queiroz Viana Ferreira, Karina Andrea Gómez, Fred Luciano Neves Santos.

**Methodology:** Micaela Soledad Ossowski, Tycha Bianca Sabaini Pavan, Randrin Queiroz Viana Ferreira, Felipe Silva Santos de Jesus, Fred Luciano Neves Santos.

**Project administration:** Karina Andrea Gómez, Fred Luciano Neves Santos.

**Resources:** Fred Luciano Neves Santos.

**Software:** Fred Luciano Neves Santos.

**Supervision:** Fred Luciano Neves Santos.

**Validation:** Larissa de Carvalho Medrado Vasconcelos, Micaela Soledad Ossowski, Daniel Dias Sampaio, Felipe Silva Santos de Jesus, Karina Andrea Gómez, Fred Luciano Neves Santos.

**Visualization:** Larissa de Carvalho Medrado Vasconcelos, Daniel Dias Sampaio, Fred Luciano Neves Santos.

**Writing – original draft:** Larissa de Carvalho Medrado Vasconcelos, Micaela Soledad Ossowski, Edimilson Domingos Silva, Daniel Dias Sampaio, Tycha Bianca Sabaini Pavan, Randrin Queiroz Viana Ferreira, Felipe Silva Santos de Jesus, Raúl Chadi, Marisa Liliana Fernández, Yolanda Hernández, Fred Luciano Neves Santos.

**Writing – review & editing:** Micaela Soledad Ossowski, Karina Andrea Gómez, Fred Luciano Neves Santos.

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
