## [Decision Letter · Decision Letter 0]

23 Feb 2026

PNTD-D-26-00016

Diagnostic Performance of the TR Chagas Bio-Manguinhos Rapid Test for Detecting Anti-Trypanosomacruzi IgG in Human Samples from three Southern Cone countries

Dear Dr. Santos,

Thank you for submitting your manuscript to PLOS Neglected Tropical Diseases. After careful consideration, we feel that it has merit but does not fully meet PLOS Neglected Tropical Diseases's publication criteria as it currently stands. Therefore, we invite you to submit a revised version of the manuscript that addresses the points raised during the review process.

Please submit your revised manuscript within by Apr 24 2026 11:59PM. If you will need more time than this to complete your revisions, please reply to this message or contact the journal office at plosntds@plos.org. Please include the following items when submitting your revised manuscript:

We look forward to receiving your revised manuscript.

Kind regards,

Pierre Buekens

Academic Editor

Susan Madison-Antenucci

Section Editor

Shaden Kamhawi

co-Editor-in-Chief

Paul Brindley

co-Editor-in-Chief

**Additional Editor Comments:**

Please take the thoughtful and detailed comments and suggestions of the reviewers into account.

**Journal Requirements:**

1) Some material included in your submission may be copyrighted. According to PLOSu2019s copyright policy, authors who use figures or other material (e.g., graphics, clipart, maps) from another author or copyright holder must demonstrate or obtain permission to publish this material under the Creative Commons Attribution 4.0 International (CC BY 4.0) License used by PLOS journals. Please closely review the details of PLOSu2019s copyright requirements here: PLOS Licenses and Copyright. If you need to request permissions from a copyright holder, you may use PLOS's Copyright Content Permission form.

Potential Copyright Issues:

i) Figure 1. Please (a) provide a direct link to the base layer of the map (i.e., the country or region border shape) and ensure this is also included in the figure legend; and (b) provide a link to the terms of use / license information for the base layer image or shapefile. We cannot publish proprietary or copyrighted maps (e.g. Google Maps, Mapquest) and the terms of use for your map base layer must be compatible with our CC BY 4.0 license.

2) Please ensure that the funders and grant numbers match between the Financial Disclosure field and the Funding Information tab in your submission form. Note that the funders must be provided in the same order in both places as well.

**Reviewers' Comments:**

Reviewer's Responses to Questions

**Key Review Criteria Required for Acceptance?**

**Methods**

-Are the objectives of the study clearly articulated with a clear testable hypothesis stated?

-Is the study design appropriate to address the stated objectives?

-Is the population clearly described and appropriate for the hypothesis being tested?

-Is the sample size sufficient to ensure adequate power to address the hypothesis being tested?

-Were correct statistical analysis used to support conclusions?

-Are there concerns about ethical or regulatory requirements being met?

Reviewer #1: The study’s methodology is sound, clearly presented and the work adheres to ethical standards.

A few minor issues should be addressed to further strengthen the manuscript.

I hope these comments are helpful.

Study Design and Samples

• Please indicate whether the study protocol is publicly available or can be shared upon request. If applicable, stating this explicitly would enhance transparency.

• Please specify when and where the samples were collected (study sites and months/years), as well as the time interval between the index test and the reference standard.

• Given that samples were stored for several months or years, please clarify how sample quality was ensured. Were samples re-tested using the reference standard, or were reference results derived from the original study? A brief discussion of how this may influence interpretation of the results would be helpful.

• Please clarify whether commercial assays (eg. ELISA, HAI) or in-house methods were used as part of the composite reference standard, and include references as appropriate.

• The sample size estimation would benefit from additional clarification. Please specify the power used, whether the reported margin of error (1.7) corresponds to a half-width or full-width interval, and provide a reference for the formula applied.

For context, estimating 99% sensitivity with a 3.5% full-width margin of error, 95% confidence, and 80% power requires approximately 254 confirmed cases (and a similar number of negative cases for 99% specificity), according to (Zhou, X. H., McClish, D. K., & Obuchowski, N. A. (2009). Statistical methods in diagnostic medicine. John Wiley & Sons). This may help explain the uncertainty observed, particularly for specificity.

• Please clarify whether an intended sample size was defined for stratified analyses (by country and cardiac involvement), and how this was determined.

• Please indicate whether any External Quality Assessment (EQA) was conducted. If not, a brief discussion of this point would still be informative. Use of a serological EQA panel aligned with international or national standards could strengthen confidence in test performance and observer consistency.

• Please clarify the blinding procedures applied in the study. Specifically, were observers blinded to clinical information, reference test results, and each other’s interpretations? This information is important for assessing the potential risk of review bias. If full blinding was not feasible, please discuss any potential impact on the findings.

Reviewer #2: This manuscript presents an evaluation of the TR Chagas Bio-Manguinhos rapid diagnostic test in samples from Argentina, Bolivia and Paraguay. The topic is relevant for Chagas disease diagnostics in Latin America. My comment below focus on methodology and interpretation of the results.

Major comments: Methods

Study design and samples:

How was T. cruzi infection status defined? How many assays were used?

Were samples selected consecutively or convenience samples from the repository? Could spectrum bias affect test performance estimates?

Line 133: Any samples with discordant results? PCR test performed.

Line 140: Please specify which third confirmatory assay was used to resolve discordant results (manufacturer, platform, antigen target, cutoff criteria).

How many samples showed discordant results between the index test and the reference standard?

Were discordant samples reclassified based solely on the third test, or using a composite algorithm?

The supplementary table currently reports only final positive/negative serological profiles. Could the authors include a breakdown showing discordant cases and how they were resolved?

Reference standard:

The reference standard used to classify samples as T. cruzi–positive or negative is not clearly and explicitly defined. The specific assays included in the composite reference standard and the decision algorithm applied should be described in detail.

Handling of discordant results:

The Methods state that discordant serological results were resolved using a third confirmatory test; however, the specific assay used, the number of discordant samples, and how these cases were incorporated into the final analysis are not reported.

Antigen composition of the test:

The antigenic composition of the TR Chagas Bio-Manguinhos rapid test is not described in the Methods section and is only mentioned in the discussion. For transparency and reproducibility, this information should be clearly stated in the Methods.

Sample size and subgroup analyses:

While sample size calculations are provided, additional clarification is needed regarding the assumptions used and how the final sample size relates to the stratified analyses by geography and clinical stage.

Sample selection and representativeness:

Additional details on sample selection, geographical coverage, and clinical spectrum would help contextualize the findings, particularly given the use of archived plasma samples and the absence of province-level data.

Statistical analysis:

The statistical methods are generally appropriate; however, the limited power of stratified analyses, as mentioned, should be acknowledged when interpreting subgroup comparisons.

Reviewer #3: Although the Introduction states that “In this context, we evaluated the diagnostic performance of the TR Chagas Bio‑Manguinhos test using serum samples from Argentina, Bolivia, and Paraguay, applying international guidelines for diagnostic test validation,” the subsequent Methods and Results sections introduce additional analyses related to the country of origin of the samples and the cardiological involvement of the patients. These important analytical dimensions are not mentioned or anticipated in the Introduction. For coherence and to guide the reader, the Introduction should briefly indicate that the study also examines potential differences by sample origin and by clinical/cardiovascular status.

The study design is appropriate, and the target population is well defined.

Ethical approvals are adequate.

The analytical tools employed are suitable. However, the authors do not clearly demonstrate whether the sample size is statistically powered to support stratified analyses based on country of origin or cardiological involvement.

1. Line 129 — Study design clarity

For readers’ comprehension, please state explicitly that the study is retrospective and observational, and that samples were obtained from an INGEBI-CONICET serotheque/biorepository.

2. Line 133 — Sample size and stratification

In addition to describing the sample size calculation, please specify whether the resulting N supports stratified analyses by country of origin and by clinical/cardiovascular classification.

Include the biostatistical approach and bibliographic reference for calculating sample size

3. Line 134 — Not multicenter

For clarity, please state that although samples originate from multiple countries, this is not a multicenter study; it is a single center evaluation using serum samples from people from three countries.

4. Line 136 — Clinical evaluation and sampling

The text mentions medical evaluation. Because this is an RDT performance assessment, please also describe the sample collection process (collection setting and procedures) in addition to clinical evaluation.

5. Line 137 — Sample metadata and matrix

In the Samples section, please include: dates of collection, storage conditions (temperature, duration, freeze–thaw history), and explicitly state the biological matrix (plasma).

6. RDT reading procedures

Please detail the reading procedures for RDT:

• Number of operators (one vs. two readers; if discordant, was there a third adjudicator?).

• Blinding: confirm that operators/readers were blinded to conventional serology results and any clinical data.

• Training/competency of operators and any standardized reading time window and lighting conditions, if applicable.

**Results**

-Does the analysis presented match the analysis plan?

-Are the results clearly and completely presented?

-Are the figures (Tables, Images) of sufficient quality for clarity?

Reviewer #1: Results are clearly presented and supported by appropriate analyses. A few minor issues should be addressed to further strengthen the manuscript.

• Please consider adding p-values or clearly indicating whether comparisons are statistically significant, not only in the main text but also in the corresponding tables.

• If available, please indicate whether the distribution of alternative diagnoses is known among samples classified as negative.

Reviewer #2: Although this study is presented as the first evaluation of this assay in this region, the manuscript does not describe the antigenic targets or peptide composition of the test. Additional detail on antigen composition would help readers place this assay within the broader landscape of existing Chagas serological tests.

Table 1. The authors state that diagnostic performance was evaluated against a reference standard; however, Table 1 does not specify which reference standard was used. Because sensitivity, specificity, likelihood ratios, and agreement metrics are entirely dependent on the reference standard, this information is essential for interpretation and reproducibility. Please clearly identify the reference standard used (including assays and decision algorithm) in both the Methods and Table 1.

Table 2. The authors state that “all 21 Paraguayan-born individuals with confirmed infection were correctly classified, resulting in a sensitivity of 100% (95% CI: 84.5%–100%).” However, in Table 2, the total number of Paraguayan-born individuals is reported as n=8, which is inconsistent with the reported number of true positives (TP=21). Please revise Table 2 and ensure that the total number of samples, true positives, and false negatives are correctly and consistently reported.

Iine 269: The number of samples in advanced stages, particularly stage 2 (n=8) is very small, resulting in wide confidence intervals and limited statistical power. Therefore, the absence of statistically significant differences is more likely due to limited sample size than true equivalence of sensitivity across stages. From a serological standpoint, this analysis should be viewed as exploratory or taken with caution.

Reviewer #3: 1. Lines 266–269 — Stratified analyses feasibility

Please clarify whether the sample size supports stratified analyses for the outcomes reported (e.g., by country of origin and by clinical/cardiovascular involvement), and present these analyses if adequately powered—or justify why not.

2. Line 235 onwards : Specificity analysis

Please explain why specificity was not analyzed and only sensitivity was reported in geographical origen and cardiac involvement. Adjust it for a complete analysis

**Conclusions**

-Are the conclusions supported by the data presented?

-Are the limitations of analysis clearly described?

-Do the authors discuss how these data can be helpful to advance our understanding of the topic under study?

-Is public health relevance addressed?

Reviewer #1: Conclusions are supported by the data presented. The authors commented about some of the study limitations, and compared their results with previous studies. There are a few points that should be addressed to improve the clarity and robustness.

Discussion

• Please clarify whether the test configuration evaluated (IBMP-8.1 and IBMP-8.4 antigens) corresponds to the same product version used in previous evaluations.

• Additional discussion of study limitations, including potential sources of bias and statistical uncertainty, would strengthen interpretation of the findings, particularly regarding uncertainty in specificity estimates, the timing between index and reference testing, and possible effects of long-term sample storage.

• Discussion of the current regulatory status and use of the test (in Brazil only?), as well as its potential relevance for market entry in other countries, would add value. Given the unique position of Bio-Manguinhos as a public manufacturer in a fragmented market, this discussion could be particularly informative in the context of access in underserved populations and affordability.

Acknowledgements

• Please clarify the role of the test manufacturer/developer (Bio-Manguinhos) in the study. Specifically, indicate whether the tests were donated and whether the manufacturer had any role in study design, supervision, data analysis, or interpretation.

Reviewer #2: Discussion

Line 296: It is unclear how the authors can conclude that the TR Chagas Bio-Manguinhos test matched or exceeded the sensitivity of other LFAs evaluated in multicountry comparisons, as this study did not include samples from a wide range of countries with known differences in circulating DTUs or parasite strains, nor direct head-to-head comparisons.

Line 331: Given the limited sample sizes and wide confidence intervals in several stratified analyses. Please clarify

Major comment

The manuscript cites prior work using IBMP-based chimeric antigens; however, the antigenic composition of the TR Chagas Bio-Manguinhos assay evaluated here is not clearly described. Neither the present study nor the referenced work provides sufficient detail regarding the specific antigens or peptides/epitopes included in the test.

The description of antigen composition would facilitate interpretation of the biological basis of the assay and help readers better contextualize this platform relative to existing commercial tests (e,g., Wiener, InBios). Given that this is presented as the first evaluation of the assay in this region, such clarification is important to support interpretation of the findings.

Reviewer #3: The Discussion and Conclusions are clearly articulated regarding the performance evaluation of the diagnostic test under study. However, several important points require clarification or further elaboration:

1. Line 276 — Not multicenter (clarification needed)

To avoid any misunderstanding, please state explicitly that this is not a multicenter study, but rather a single‑center evaluation using samples from patients originating from three countries.

2. Line 315 — Characterization of negatives in retrospective panels

In retrospective evaluations using pre‑characterized panels, negative samples are expected to be true negatives according to the reference standard. Therefore, statements suggesting possible cross‑reactivity with other infections (e.g., leishmaniasis) should be reconsidered or justified based on the actual composition and characterization of the negative panel used in this study. A revision of this point is recommended.

3. Line 300 — Mischaracterization of the cited study

The cited study is not multicenter; it was retrospective (using biobank/serotheque sera) and carried out under controlled laboratory conditions. The description should be corrected to accurately reflect the study design.

4. Lines 315–317 — Both sensitivity and specificity are context‑dependent

The discussion currently frames specificity as the parameter conditioned by context; however, both sensitivity and specificity depend on multiple contextual factors, including:

Characteristics of the positive samples (e.g., chronic indeterminate Chagas with lower antibody titers may reduce sensitivity in some assays).

Population‑level factors (e.g., immunosuppression, which can lead to diminished antibody production and thus lower serological sensitivity).

Strain diversity/DTUs, given that antigenic variation may influence recognition.

The reference standard used, as imperfect comparators may bias estimates of both sensitivity and specificity.

5. In the Bolivian study by López et al. (2024)( López R, García A, Chura Aruni JJ, Balboa V, Rodríguez A, Erkosar B, et al. Comparative evaluation of lateral flow assays to diagnose chronic Trypanosoma cruzi infection in Bolivia. PLoS Negl Trop Dis 18(3): e0012016 ), sensitivity values were similar to those reported here; however, specificity differed markedly despite the substantially larger sample size (N = 398; positives = 216). The authors should explore and discuss potential reasons underlying this discrepancy, considering methodological differences, sample composition, and epidemiological context.

6. Sample size considerations and discrepancies with prior evaluations

Previous evaluations in Bolivia and Colombia—where this RDT was assessed alongside several others—reported much higher specificity for the TR Bio‑Manguinhos test, using sample sizes comparable to those used for the other assays in those studies. Because those evaluations included a higher proportion of negative samples, the authors should clarify how the sample size assumptions used here address these differences and assess whether variation in sample composition may explain the lower specificity reported in the present study.

7. Implications for large‑scale implementation and cost‑effectiveness

Given the potential large‑scale use of rapid diagnostic tests for screening, assays with high sensitivity but more limited specificity could substantially increase downstream diagnostic confirmation costs. The authors should consider recommending cost‑effectiveness analyses to evaluate the feasibility of deploying this RDT in broad screening strategies.

8. Need for discussion on confirmatory use and public health impact

The Discussion section does not explore future directions for applied research regarding the potential use of RDTs as confirmatory diagnostic tools, nor does it address the broader public health implications should such applications prove viable. Expanding this section would significantly strengthen the translational and programmatic relevance of the manuscript.

**Editorial and Data Presentation Modifications?**

Reviewer #1: • Throughout the manuscript, please consider avoiding terms such as “chronic CD,” “CD screening” or “cases of CD” In many instances, more precise terminology would be “detection of chronic T. cruzi infection” “screening for T. cruzi infection” or “individuals infected with T. cruzi” as not all infected individuals develop clinical disease.

Study Design and Samples

• Please clarify whether photographs of the evaluated index test were taken and whether they could be made available upon request.

• Please clarify whether discordant interpretations resolved by a third observer were adjudicated directly from the cassette or using stored images.

• If available, reporting inter-observer agreement and the rate of invalid test results would further strengthen the results.

Discussion

• The paragraph spanning lines 298–322 could be shortened or divided into two parts to improve readability.

• Please note that the study by Marchiol et al. is not a multicenter study; revising this wording would improve accuracy.

• Please specify which references are meant by “prior evaluations frequently included more heterogeneous negative panels”.

• The discussion would benefit from addressing potential lot-to-lot variation. If available, please indicate whether this aspect was considered in the present study or reported in previous evaluations of the index test.

Reviewer #2: The manuscript will benefit from several editorial and presentation related clarification to improve methods. In particular, the antigens used in each diagnostic assay should be explicitly stated in the Methods section and clearly linked to the corresponding results and figures, rather than being implied. Additionally, clearer descriptions of the reference standard used for test evaluation and more explicit labeling of tables and figures would improve interpretability.

Reviewer #3: See comments above

**Summary and General Comments**

Reviewer #1: It is a very interesting and well-structured study about a novel evaluation of the index test with samples from the Southern Cone. I consider it makes valuable contributions to the field of Trypanosoma cruzi infection detection. The work will be of interest to researchers, test developers and manufacturers, practitioners, and policy makers in the field of NTDs, as well as to broader audiences. The study is methodologically robust, with clearly reported findings supported by appropriate analytical approaches, and was conducted in accordance with ethical requirements.

There are a few minor issues that should be addressed to improve the clarity and robustness of the manuscript. Therefore, I recommend the manuscript for publication pending minor revisions, which are detailed above. I hope this is helpful.

Reviewer #2: In summary: The study addresses an important regional gap in Chagas diagnostics and provides useful data from an underrepresented setting. However, major revisions are required to clarify key methodological aspects, including antigen composition, definition of the reference standard, and handling of discordant samples. In addition, the interpretation of stratified analyses should be moderated, and the Discussion should be more closely aligned with the data generated in this study, particularly with respect to claims of novelty and comparative performance with existing commercial assays.

Reviewer #3: The topic addressed in this manuscript is of substantial importance for advancing the simplification of diagnostic approaches for Chagas disease. The authors successfully highlight the need to expand the use of rapid diagnostic tests while simultaneously ensuring rigorous evaluation of their quality. This is a highly relevant and timely contribution to the field, and the manuscript is therefore of considerable interest.

PLOS authors have the option to publish the peer review history of their article (what does this mean?). If published, this will include your full peer review and any attached files.

**Do you want your identity to be public for this peer review?** For information about this choice, including consent withdrawal, please see our Privacy Policy.

Reviewer #1: **Yes:**Laura C. Bohorquez

Reviewer #2: No

Reviewer #3: No

**Figure resubmission:**

After uploading your figures to PLOS’s NAAS tool - https://ngplosjournals.pagemajik.ai/artanalysis, NAAS will process the files provided and display the results in the Uploaded Files section of the page as the processing is complete. If the uploaded figures meet our requirements (or NAAS is able to fix the files to meet our requirements), the figure will be marked as fixed above. If NAAS is unable to fix the files, a red failed label will appear above. When NAAS has confirmed that the figure files meet our requirements, please download the file via the download option, and include these NAAS processed figure files when submitting your revised manuscript. **Reproducibility:**

---

## [Decision Letter · Decision Letter 1]

22 Apr 2026

PNTD-D-26-00016R1

Diagnostic Performance of the TR Chagas Bio-Manguinhos Rapid Test for Detecting Anti-Trypanosomacruzi IgG in Human Samples from three Southern Cone countries

Dear Dr. Santos,

Thank you for submitting your manuscript to PLOS Neglected Tropical Diseases. After careful consideration, we feel that it has merit but does not fully meet PLOS Neglected Tropical Diseases's publication criteria as it currently stands. Therefore, we invite you to submit a revised version of the manuscript that addresses the points raised during the review process.

We look forward to receiving your revised manuscript.

Kind regards,

Pierre Buekens

Academic Editor

Susan Madison-Antenucci

Section Editor

Shaden Kamhawi

co-Editor-in-Chief

Paul Brindley

co-Editor-in-Chief

**Additional Editor Comments:**

Please further expand the discussion on the exclusion of discordant samples, particularly regarding its potential impact on diagnostic performance estimates and generalizability.

**Journal Requirements:**

**Reviewers' comments:**

Reviewer's Responses to Questions

**Key Review Criteria Required for Acceptance?**

**Methods**

-Are the objectives of the study clearly articulated with a clear testable hypothesis stated?

-Is the study design appropriate to address the stated objectives?

-Is the population clearly described and appropriate for the hypothesis being tested?

-Is the sample size sufficient to ensure adequate power to address the hypothesis being tested?

-Were correct statistical analysis used to support conclusions?

-Are there concerns about ethical or regulatory requirements being met?

Reviewer #1: The authors have addressed my concerns thoroughly and the manuscript is much improved for it. I have no further comments and am happy to recommend acceptance.

Reviewer #2: Yes

Reviewer #3: (No Response)

**Results**

-Does the analysis presented match the analysis plan?

-Are the results clearly and completely presented?

-Are the figures (Tables, Images) of sufficient quality for clarity?

Reviewer #1: The authors have addressed my concerns thoroughly and the manuscript is much improved for it. I have no further comments and am happy to recommend acceptance.

Reviewer #2: Yes

Reviewer #3: (No Response)

**Conclusions**

-Are the conclusions supported by the data presented?

-Are the limitations of analysis clearly described?

-Do the authors discuss how these data can be helpful to advance our understanding of the topic under study?

-Is public health relevance addressed?

Reviewer #1: The authors have addressed my concerns thoroughly and the manuscript is much improved for it. I have no further comments and am happy to recommend acceptance.

Reviewer #2: Yes

Reviewer #3: (No Response)

**Editorial and Data Presentation Modifications?**

Reviewer #1: The authors have addressed my concerns thoroughly and the manuscript is much improved for it. I have no further comments and am happy to recommend acceptance.

Reviewer #2: yes

Reviewer #3: (No Response)

**Summary and General Comments**

Reviewer #1: Thank you for the careful and thorough revision. The authors have addressed my previous concerns in a satisfying manner, and the manuscript is substantially improved as a result.

I have no further concerns. I am pleased to recommend this manuscript for acceptance in its current form.

Reviewer #2: The authors have addressed the comments well, and the manuscript has clearly improved. I recommend minor revisions. I encourage the authors to further expand the discussion on the exclusion of discordant samples, particularly regarding its potential impact on diagnostic performance estimates and generalizability.

Reviewer #3: (No Response)

PLOS authors have the option to publish the peer review history of their article (what does this mean?). If published, this will include your full peer review and any attached files.

**Do you want your identity to be public for this peer review?** For information about this choice, including consent withdrawal, please see our Privacy Policy.

Reviewer #1: **Yes:**Laura C. Bohorquez

Reviewer #2: No

Reviewer #3: No

**Figure resubmission:**

After uploading your figures to PLOS’s NAAS tool - https://ngplosjournals.pagemajik.ai/artanalysis, NAAS will process the files provided and display the results in the Uploaded Files section of the page as the processing is complete. If the uploaded figures meet our requirements (or NAAS is able to fix the files to meet our requirements), the figure will be marked as fixed above. If NAAS is unable to fix the files, a red failed label will appear above. When NAAS has confirmed that the figure files meet our requirements, please download the file via the download option, and include these NAAS processed figure files when submitting your revised manuscript.
---

## [Editor Report · Decision Letter 2]

27 Apr 2026

Dear Dr Santos,

We are pleased to inform you that your manuscript 'Diagnostic Performance of the TR Chagas Bio-Manguinhos Rapid Test for Detecting Anti-Trypanosoma cruzi IgG in Human Samples from three Southern Cone countries' has been provisionally accepted for publication in PLOS Neglected Tropical Diseases.

Best regards,

Pierre Buekens

Academic Editor

Susan Madison-Antenucci

Section Editor

Shaden Kamhawi

co-Editor-in-Chief

Paul Brindley

co-Editor-in-Chief

Thank you for responding to all comments and suggestions.

---

## [Editor Report · Acceptance letter]

Dear Dr Santos,

We are delighted to inform you that your manuscript, "Diagnostic Performance of the TR Chagas Bio-Manguinhos Rapid Test for Detecting Anti-Trypanosoma cruzi IgG in Human Samples from three Southern Cone countries," has been formally accepted for publication in PLOS Neglected Tropical Diseases.

Best regards,

Shaden Kamhawi

co-Editor-in-Chief

Paul Brindley

co-Editor-in-Chief
